# Investigation of Spatial and Temporal Changes in the Land Surface Albedo for the Entire Chinese Territory

**Jihui Yuan** 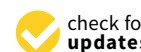

Department of Architecture and Civil Engineering, Toyohashi University of Technology, 1-1 Hibarigaoka, Tempaku-cho, Toyohashi, Aichi 441-8580, Japan; yuan@ace.tut.ac.jp; Tel.: +81-(0)-532-44-6839

**Abstract:** Currently, global climate change (GCC) and the urban heat island (UHI) phenomena are becoming serious problems, partly due to the artificial construction of the land surface. When sunlight reaches the land surface, some of it is absorbed and some is reflected. The state of the land surface directly affects the surface albedo, which determines the magnitude of solar radiation reflected by the land surface in the daytime. In order to better understand the spatial and temporal changes in surface albedo, this study investigated and analyzed the surface albedo from 2000 to 2016 (2000, 2008, and 2016) in the entire Chinese territory, based on the measurement database obtained from the Moderate Resolution Imaging Spectroradiometer (MODIS) instrument, aboard NASA's Terra satellite. It was shown that the Northeast China exhibited the largest decline in surface albedo and North China showed the largest rising trend of surface albedo from 2000 to 2016. The correlation between changes in surface albedo and the Normalized Difference Vegetation Index (NDVI) indicated that the change trend of surface albedo was opposite to that of NDVI. In addition, in order to better understand the distribution of surface albedo in the entire Chinese territory, the classifications of surface albedo in three years (2000, 2008, and 2016) were implemented using five classification methods in this study.

**Keywords:** global climate change; urban heat island; Chinese territory; surface albedo; classification; normalized difference vegetation index

---

## 1. Introduction

### 1.1. Urbanization

Urbanization refers to the population shift from rural to urban areas, the gradual increase in the proportion of people living in urban areas, and the ways in which each society adapts to the change. It is predominantly the process through which towns and cities are formed and become larger, as more people begin living and working in central areas [1]. The continuing urbanization and overall growth of the world's population is projected to add 2.5 billion people to the urban population by 2050, with nearly 90% of the increase concentrated in Asia and Africa. At the same time, the proportion of the world's population living in urban areas is expected to increase, reaching 66% by 2050 [2]. Additionally, it is predicted that by 2050, about 64% of the developing world and 86% of the developed world will be urbanized [3]. This would be equivalent to approximately 3 billion urbanites by 2050, much of which would occur in Africa and Asia [4]. Notably, the United Nations also recently projected that nearly all global population growth from 2017 to 2030 would be absorbed by cities, about 1.1 billion new urbanites over the next 13 years [5]. Urbanization in China increased in speed following the initiation of the reform and opening policy. By the end of 2015, 56% of the total population lived in urban areas, a dramatic increase from 26% in 1990 [6]. Human activities (e.g., deforestation, farming, and urbanization) change the albedo of various areas around the globe. However, apart from the remote-sensing technique, there are some simpler ways to quantify this effect on the global scale.

*1.2. Global Climate Change and Urban Heat Island*

Global climate change (GCC) and urban heat island (UHI) intensification are making cities hotter places to live. The GCC is the slow increase in the average temperature of the earth's atmosphere, because an increased amount of the energy (heat) striking the earth from the sun is currently being trapped in the atmosphere and not radiated out into space [7], and is strongly affecting the global energy balance [8]. The UHI is an urban area or metropolitan area that is significantly warmer than its surrounding rural areas, due to human activities. The temperature difference is usually larger at night than during the day, and is most apparent when winds are weak [9]. UHI is most noticeable during the summer and winter. The main cause of the UHI effect is from modification of land surfaces, urban block geometry, poor street canyon ventilation, and reduction of green areas, etc. [10]. It is shown that urban trees could not only reduce health hazards by removing pollutants from the air, but could also mitigate the UHI effect by shading the city and absorbing heat under the action of latent heat evaporation [11]. Waste heat generated by energy use is a secondary contributor [12]. The UHI phenomenon was studied for a long time by many scholars worldwide. Howard [13] discovered the impact of London upon its climate by comparing his temperature records against those made by the Royal Society at the Somerset House. His discovery indicated that the temperature of the city should not be regarded as the climate temperature under normal circumstances, because the temperature of the city was strongly affected by its structure, crowded population, and energy consumption, such as fuel. Bornstein [14] used helicopter to investigate the UHI phenomenon of New York, USA, and his results showed that the UHI phenomenon was the most intense near the ground and almost decreased to zero at a height of about 300 m. Saitoh et al. [15] developed 3D models to examine the effects of UHI phenomenon in Tokyo, Japan, by using satellite and land survey data. Kato and Yamaguchi [16] used Landsat and ASTER data to analyze the seasonal changes pertaining to UHI in Nagoya, Japan. A study by Kim [17] reported that the surface temperature of the city center was higher than the surrounding vegetative areas, by up to 10 °C in Washington.

*1.3. Urban Albedo and Satellite Technology*

Albedo is a measure for reflectivity or optical brightness of the land surface. Land surface albedo is defined as the ratio of irradiance reflected to the irradiance received by a surface [18]. The proportion reflected is not only determined by properties of the surface itself, but also by the spectral and angular distribution of solar radiation reaching the land surface [19]. Albedo is an important concept in climatology, astronomy, and environmental management (e.g., as part of the Leadership in Energy and Environmental Design (LEED) program for sustainable rating of buildings), it affects climate by determining how much solar radiation the land surface absorbs [20]. The average albedo of the Earth is often affected by the observation height, and it was shown that average albedo was 0.30–0.35 at the top of the atmosphere, because of cloud cover, but widely varied locally across the surface because of different geological and environmental features [21].

The change in urban albedo was mainly due to rapid urbanization on the urban radiation budget [22]. Hu et al. [23] used a modified radiative forcing (RF) derivation approach, based on Landsat images of satellite technology to quantify changes in the solar radiation budget, induced by variations in surface albedo in Beijing from 2001 to 2009. The results showed that there was rapid urban expansion over the last decade, with the urban land area increasing at about 3.3% annually from 2001 to 2009, resulting in a lower albedo, due to the complex building configurations of urban centers and the higher albedo on flat surfaces of suburban areas and cropland. Hou et al. [24] used a fine resolution green vegetation fraction (GVF) and surface albedo values derived from Landsat Thematic Mapper/Enhanced Thematic Mapper Plus images from 1990 to 2000, to examine the relationship of both GVF and surface albedo values to the spatial gradients of parameters related to dramatic urbanization in South China. The results showed that vegetation cover modified by urban expansion changed the surface reflectivity and influenced the surface energy balance. Many studies indicated that urban

temperatures and photochemical reaction rates increased with the increase in urban albedo [25,26], and increasing urban albedo was regarded as a verifiable and repeatable UHI mitigation strategy [27,28].

### 1.4. Aims of This Study

Surface albedo has become an important issue, because it is considered that surface albedo is strongly related to the UHI phenomenon mitigation and energy conservation [29]. Moreover, surface albedo and its temporal change was also investigated by using optical satellite data from the Advanced Very High Resolution Radiometer (AVHRR), drone observation data, etc. [30].

Compared to the raster maps, many studies used contour maps for better reflection of spatial and temporal variations. Ahmadi and Sedghamiz [31] and Machiwal et al. [32] used contour maps for geostatistical analysis of spatial and temporal variations of groundwater level. The above-mentioned studies used advanced image analysis technology and drone observation technology, and were limited to a certain areas such as by Hu et al. [23]. On the contrary, this study proposed the use of a simple technology of drawing contour maps to investigate and analyze the change in surface albedo of China from 2000 to 2016, to better understand the spatial and temporal variations of surface albedo in the entire Chinese territory, based on the measurement database from the MODIS instrument aboard NASA's Terra satellite. In addition, the surface albedo in the entire Chinese territory in different years was classified using different classification methods, and the correlation between surface albedo and Normalized Difference Vegetation Index (NDVI) was also analyzed in this study.

## 2. Methodology and Materials

### 2.1. Study Area

In this study, we chose the entire Chinese territory as the study area. The territory of China lies between latitudes 18° and 54° N, and longitudes 73° and 135° E. The Chinese territory consists of 23 provinces, 5 autonomous regions, 4 municipalities, and 2 special administrative regions (as shown in Figure 1 [33]).

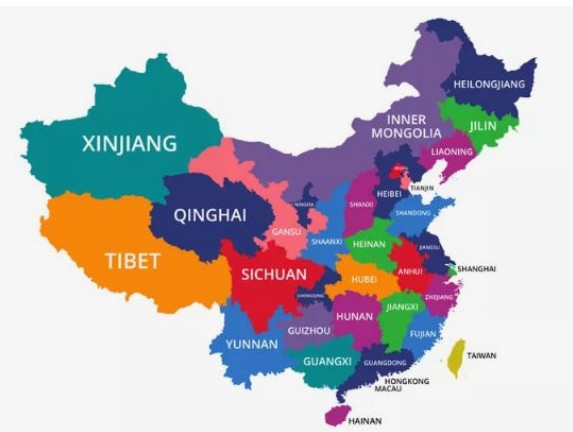

**Figure 1.** Map of the Chinese territory [33].

### 2.2. Database

Albedo is the directional hemispherical reflectance that is obtained by integrating the bi-directional reflectance distribution function (BRDF) over the exitance hemisphere (i.e., from horizon to horizon) for a single irradiance direction; in other words, as if all light was coming from one direction. The observations of albedo are based on atmospherically corrected, cloud-cleared reflectance observations from the MODIS sensors on NASA's Aqua and Terra satellites [34].

As the ice or snow of Earth's surface in winter would strongly affect the surface albedo, the albedo database (csv file type) of August from 2000 to 2016 was selected for analysis in this study. The MODIS resolution was 3600 × 1800, with 0.1 degrees in this study.

## 2.3. Method of Drawing Contour Maps

This study used a satellite observation database and a drawing software to map the surface albedo of the entire Chinese territory. The software "Surfer 14" [35], which is often used to create a contour map and a 3D map, was adopted to create the contour maps of surface albedo for the entire Chinese territory in this study.

The process of creating a contour map is detailed in the following two steps:

- 1st step: To obtain the database of surface albedo from the MODIS sensors, within the Chinese territory;
- 2nd step: To grid the surface albedo data format (.csv) into another data format (.grd), using Surfer 14;
- 3rd step: To get the polygon file of a country using "maptools" in R programming [36];
- 4th step: To add the new surface albedo data format (.grd) to the polygon file of a country (e.g., CHN.shp, CHN.bln, etc.).

## 2.4. Method of Classification

To better understand the classification of the surface albedo of the entire Chinese territory in different years, five classification methods that are most frequently used—excepting the contour maps of surface albedo—were adopted to cluster the surface albedo using the operating "classification" in R [37]. These methods included—complete-linkage clustering, median-linkage clustering, single-linkage clustering, Ward.D linkage clustering, and centroid-linkage clustering. The classified surface albedo data were then applied to draw the classification maps of surface albedo in this study.

The five main classification methods are detailed as follows.

Complete-linkage clustering is the link between the two clusters that contains all element pairs, and the distance between the clusters equals the distance between these two elements (one in each cluster) that is farthest away from each other.

The complete linkage function is shown in Equation (1),

$$D(X,Y) = max\ d(x,y) \tag{1}$$

where $D(x,y)$ is the distance between clusters $X$ and $Y$; $d(x,y)$ is the distance between elements x € X and y € Y.

The median-linkage clustering is the mean distance between the elements of each cluster and its function is shown in the Equation (2),

$$D(X,Y) = \sum\sum d(x,y)/|X|\cdot|Y| \tag{2}$$

The single-linkage clustering is the nearest distance between the elements of each cluster and its function is shown in the Equation (3),

$$D(X,Y) = min\ d(x,y) \tag{3}$$

The Ward.D-linkage clustering is also called "Ward's minimum variance method". Its function is shown in Equation (4),

$$D(X,Y) = min\ (|d(x,y)|^2) \tag{4}$$

The centroid-linkage clustering is the distance between the centroids of clusters *X* and *Y*. Its function is shown in Equation (5),

$$D(X,Y) = d \ (|CX\text{-}CY|) \tag{5}$$

where *CX* is the centroid of cluster *X*; and *CY* is the centroid of cluster *Y*.

*2.5. Normalized Difference Vegetation Index*

In order to investigate the change in the surface albedo from the side, the change in Normalized Difference Vegetation Index (NDVI), which was strongly related to surface albedo, was analyzed in this study. NDVI could quantify vegetation by measuring the difference between near-infrared (NIR) (which the vegetation strongly reflects) and red light (*R*) (which the vegetation absorbs). It always ranged from −1 to +1. When NDVI was a negative value, it was highly likely that it was water. On the other hand, if the value of NDVI was close to −1, there was a high possibility that it was dense green leaves. However, when NDVI was close to −1, there were no green leaves and it could even be an urbanized area [38]. NDVI uses the NIR and R channels in its formula, as shown in Equation (6).

$$NDVI = (NIR - R)/(NIR + R) \tag{6}$$

where *NIR* and *R* are the reflectance values in the near-infrared and red wavebands, respectively.

**3. Results**

*3.1. Contour Maps of Surface Albedo*

The distributions of surface albedo in the entire Chinese territory in different years (2000, 2008 and 2016) were created into contour maps (see Figure 2), using the Surfer 14 map tool, by inputting the surface albedo datasets obtained from the MODIS sensors.

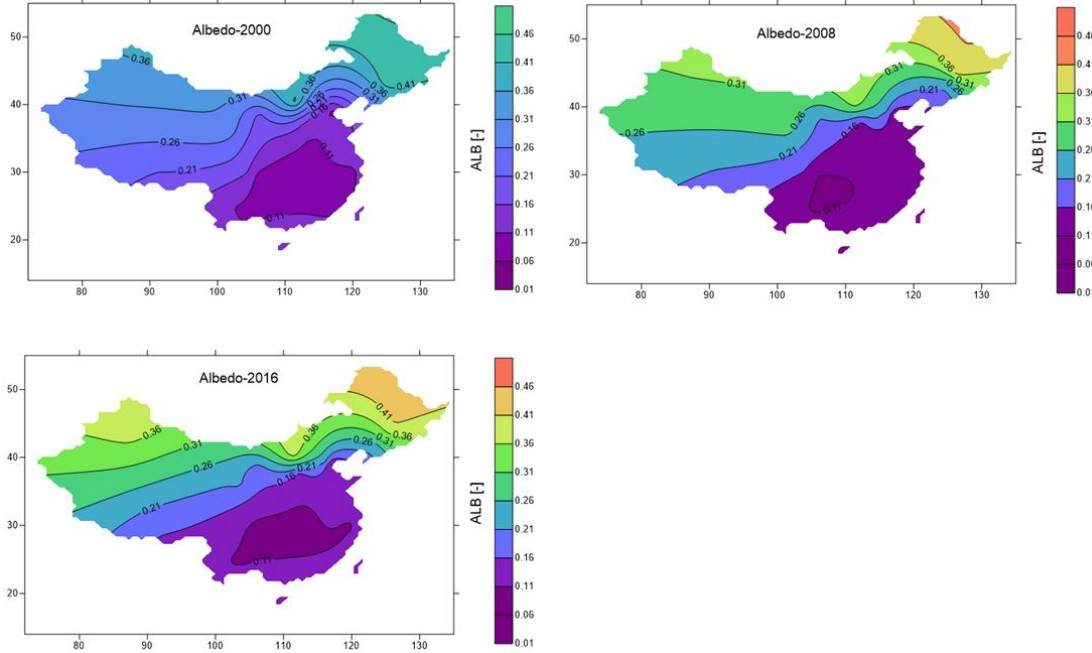

**Figure 2.** Contour maps of surface albedo in the entire Chinese territory.

Figure 2 shows a slight downward trend of surface albedo from 2000 to 2008, and then a slight upward trend from 2008 to 2016.

### 3.2. Classification of Surface Albedo

The classifications of surface albedo in different years (2000, 2008, and 2016) were mapped using five different classification methods and are shown in Figures 3–5, respectively.

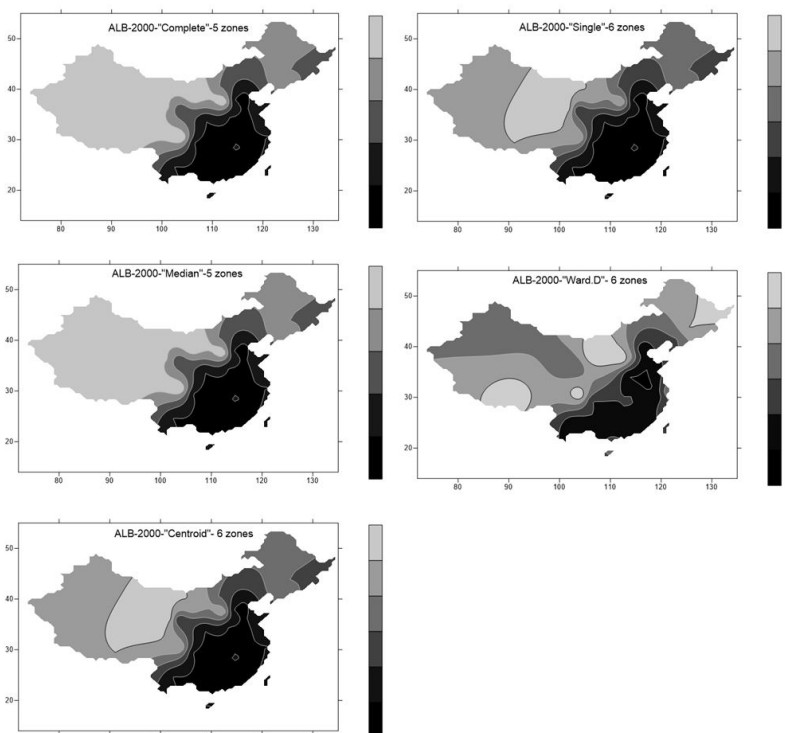

**Figure 3.** Classification of surface albedo (ALB) using 5 different methods (2000).

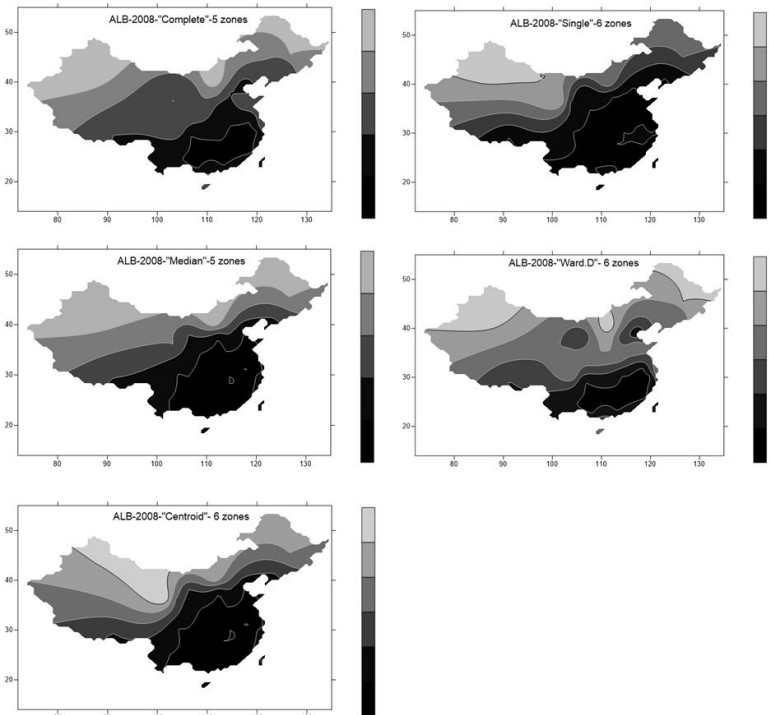

**Figure 4.** Classification of surface albedo (ALB) using 5 different methods (2008).

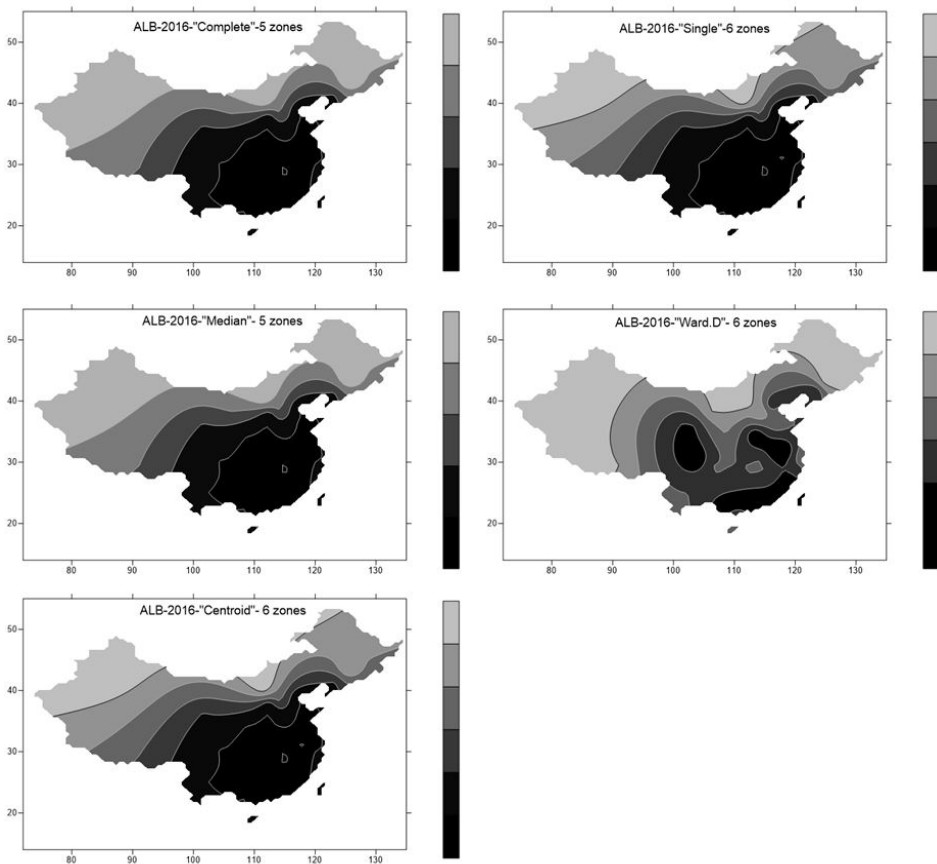

**Figure 5.** Classification of surface albedo (ALB) using 5 different methods (2016).

It was found that surface albedo could be classified into 5 zones, using the complete-linkage clustering method and the median-linkage clustering method, and was classified into 6 zones by single-linkage clustering method, Ward.D-linkage clustering method, and the centroid-linkage clustering method. Moreover, it was also found that the results obtained from the complete-linkage clustering method and the median-linkage clustering method were similar, as was that obtained by the single-linkage clustering method and the centroid-linkage clustering method.

### 3.3. Surface Albedo of Provincial Capitals

The surface albedo of all provincial capitals of China in different years (2000, 2008, and 2016) are investigated and detailed in Table 1.

**Table 1.** Surface albedo of all provincial capitals and albedo variation from 2000 to 2016.

| Provinces | Capital Cities | Location (Lat., Long.) (N, E) | ALB '00 (-) | ALB '08 (-) | ALB '16 (-) | ΔALB ('08-'00) | ΔALB ('16-'08) | ΔALB ('16-'00) |
|---|---|---|---|---|---|---|---|---|
| Anhui | Hefei | 31.5°, 117.2° | 0.11 | 0.13 | 0.12 | 0.02 | −0.01 | 0.01 |
| Beijing | Beijing | 39.5°, 116.2° | 0.13 | 0.22 | 0.22 | 0.08 | 0.00 | 0.09 |
| Chongqing | Chongqing | 29.4°, 106.3° | 0.09 | 0.10 | 0.09 | 0.01 | −0.01 | 0.00 |
| Fujian | Fuzhou | 26.1°, 119.2° | 0.11 | 0.12 | 0.12 | 0.01 | 0.00 | 0.01 |
| Gansu | Lanzhou | 36.0°, 103.5° | 0.24 | 0.24 | 0.22 | 0.00 | −0.02 | −0.02 |
| Guangdong | Guangzhou | 23.1°, 113.1° | 0.11 | 0.12 | 0.12 | 0.01 | 0.00 | 0.00 |
| Guangxi | Nanning | 22.5°, 108.2° | 0.12 | 0.12 | 0.12 | 0.01 | 0.00 | 0.00 |
| Guizhou | Guiyang | 26.4°, 106.4° | 0.10 | 0.10 | 0.11 | 0.01 | 0.00 | 0.01 |
| Hainan | Haikou | 20.0°, 110.2° | 0.14 | 0.14 | 0.14 | 0.00 | 0.00 | 0.00 |
| Hebei | Shijiazhuang | 38.0°, 114.3° | 0.14 | 0.21 | 0.19 | 0.06 | −0.01 | 0.05 |
| Heilongjiang | Harbin | 45.4°, 126.4° | 0.44 | 0.39 | 0.42 | −0.04 | 0.02 | −0.02 |
| Henan | Zhengzhou | 34.5°, 113.4° | 0.11 | 0.13 | 0.12 | 0.03 | −0.02 | 0.01 |
| Hubei | Wuhan | 30.4°, 114.2° | 0.10 | 0.12 | 0.11 | 0.02 | −0.01 | 0.01 |
| Hunan | Changsha | 28.1°, 112.6° | 0.09 | 0.11 | 0.11 | 0.02 | −0.01 | 0.01 |
| Jiangsu | Nanjing | 32.0°, 118.5° | 0.12 | 0.12 | 0.12 | −0.01 | 0.00 | 0.00 |

**Table 1.** *Cont.*

| Provinces | Capital Cities | Location (Lat., Long.) (N, E) | ALB '00 (-) | ALB '08 (-) | ALB '16 (-) | ΔALB ('08-'00) | ΔALB ('16-'08) | ΔALB ('16-'00) |
|---|---|---|---|---|---|---|---|---|
| Jiangxi | Nanchang | 28.4°, 115.6° | 0.10 | 0.11 | 0.11 | 0.01 | −0.01 | 0.01 |
| Jilin | Changchun | 43.5°, 125.2° | 0.44 | 0.24 | 0.36 | −0.20 | 0.12 | −0.09 |
| Liaoning | Shenyang | 41.5°, 123.3° | 0.32 | 0.18 | 0.23 | −0.15 | 0.06 | −0.09 |
| Inner Mongolia | Hohhot | 40.5°, 111.4° | 0.42 | 0.33 | 0.39 | −0.09 | 0.05 | −0.03 |
| Ningxia | Yinchuan | 38.3°, 106.2° | 0.20 | 0.21 | 0.19 | 0.01 | −0.02 | −0.01 |
| Paracel Islands | Paracel Islands | 16.4°, 112.0° | 0.12 | 0.15 | 0.13 | 0.02 | −0.01 | 0.01 |
| Qinghai | Xining | 36.4°, 101.5° | 0.29 | 0.27 | 0.22 | −0.02 | −0.05 | −0.07 |
| Shaanxi | Taiyuan | 37.5°, 112.3° | 0.18 | 0.14 | 0.14 | −0.04 | 0.00 | −0.03 |
| Shandong | Jinan | 36.4°, 117.0° | 0.13 | 0.14 | 0.14 | 0.01 | −0.01 | 0.00 |
| Shanghai | Shanghai | 31.1°, 121.3° | 0.15 | 0.15 | 0.15 | 0.01 | −0.01 | 0.00 |
| Shanxi | Xian | 34.2°, 108.6° | 0.14 | 0.16 | 0.14 | 0.01 | −0.02 | −0.01 |
| Sichuan | Chengdu | 30.4°, 104.0° | 0.18 | 0.16 | 0.12 | −0.02 | −0.03 | −0.06 |
| Tianjin | Tianjin | 39.0°, 117.1° | 0.10 | 0.12 | 0.11 | 0.02 | −0.02 | 0.00 |
| Xinjiang | Urumqi | 43.5°, 87.4° | 0.34 | 0.31 | 0.38 | −0.03 | 0.07 | 0.04 |
| Tibet | Lhasa | 29.4°, 91.1° | 0.20 | 0.20 | 0.17 | −0.01 | −0.03 | −0.04 |
| Yunnan | Kunming | 25.0°, 102.4° | 0.11 | 0.11 | 0.11 | 0.00 | 0.00 | 0.00 |
| Zhejiang | Hangzhou | 30.2°, 120.1° | 0.10 | 0.11 | 0.11 | 0.01 | −0.01 | 0.01 |
| | Mean | | 0.177 | 0.171 | 0.173 | −0.006 | 0.002 | −0.004 |

(ALB = Albedo; ΔALB = Albedo variation).

The surface albedo in the Northeast, West, and Northwest Chinese regions was relatively higher than that of other regions, and the average surface albedo of the whole territory was 0.177 in 2000, 0.171 in 2008, and 0.173 in 2016.

### 3.4. Surface Albedo Variations

Contour maps of surface albedo variations in the entire Chinese territory in different years (2008–2000, 2016–2008, 2016–2000) were created; shown in Figure 6. Furthermore, the surface albedo variations of all provincial capitals in different years (2008–2000, 2016–2008, 2016–2000) were also investigated and are shown in Table 1.

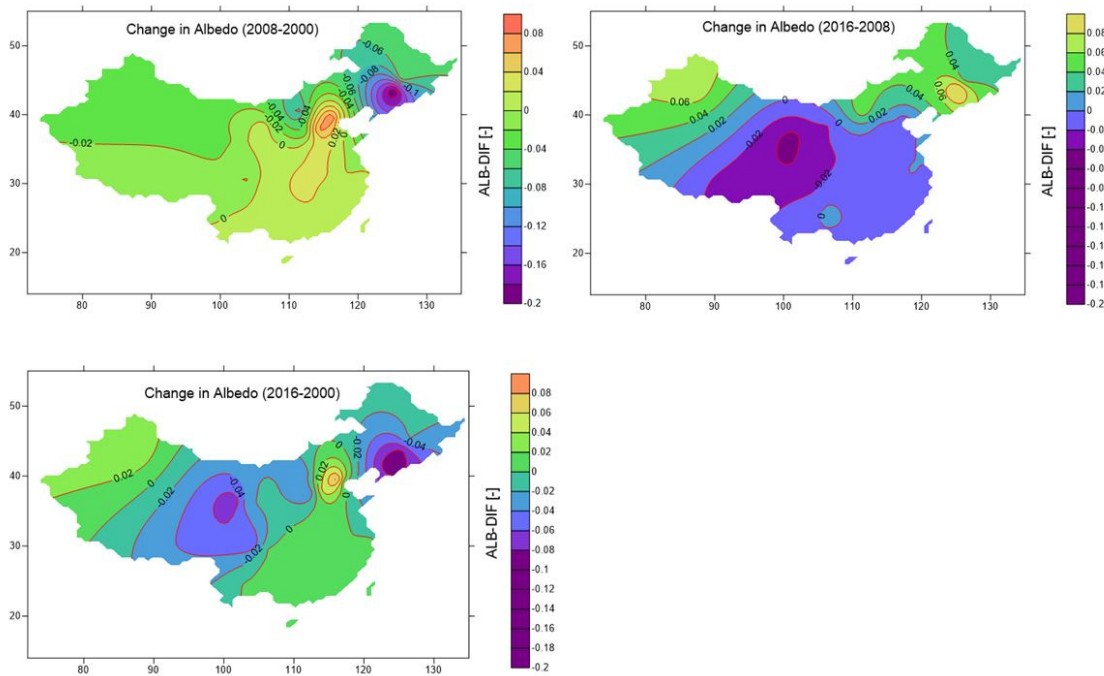

**Figure 6.** Surface albedo variations from 2000 to 2016.

Comparing the surface albedo in 2008 to that in 2000, it was found to decrease in the Northeast, West, and Northwest China. The surface albedo decreased mostly in Jilin, to about 0.20, followed by Liaoning, where it decreased about 0.15. Both Heilongjiang and Shaanxi showed a decrease of about 0.04. However, the surface albedo increased in most Chinese territory, and it increased most in Beijing,

to about 0.08, followed by Hebei, where it increased to about 0.06. In addition to the above-mentioned regions, the average of surface albedo in the other regions of China increased to about 0.01.

Comparing the surface albedo in 2016 to that in 2008, it increased in the Northeast and part of Northwest China. The surface albedo increased mostly in Jilin, to about 0.12, followed by Xinjiang, where an increase of about 0.07 was observed. Both Liaoning and Inner Mongolia showed an increase of about 0.06. However, Qinghai showed the most decrease of about 0.05 among all regions, followed by Tibet, to a decrease of about 0.03.

Comparing the surface albedo in 2016 to that in 2000, it showed a decrease in the Northeast, and parts of the Northwest and Southwest China. The surface albedo decreased most in both Jilin and Liaoning, to about 0.09, followed by Qinghai, where it decreased to about 0.07, and to about 0.06 in Sichuan. However, the surface albedo increased mostly in Beijing, to about 0.09, followed by an increase of 0.05 in Hebei, and an increase of about 0.04 in Xinjiang.

## 3.5. Change in NDVI

The data of NDVI were obtained from the MODIS, aboard NASA's Terra satellite and roughly presented the NDVI of the entire Chinese territory for different years (2000, 2008, and 2016), using the R map tool (as shown in Figure 7), excluding Hong Kong, Macao, and Taiwan. The NDVI of all provincial capitals in different years (2000, 2008, and 2016) are detailed in Table 2.

In order to investigate the correlation between surface albedo and NDVI, the change in surface albedo and NDVI from 2000 to 2016 for all provincial capitals were analyzed; the correlation is shown in Figure 8. The result indicated that the change trend of surface albedo was opposite to that of NDVI, and the correlation coefficient $R^2$ was as high as 0.9756.

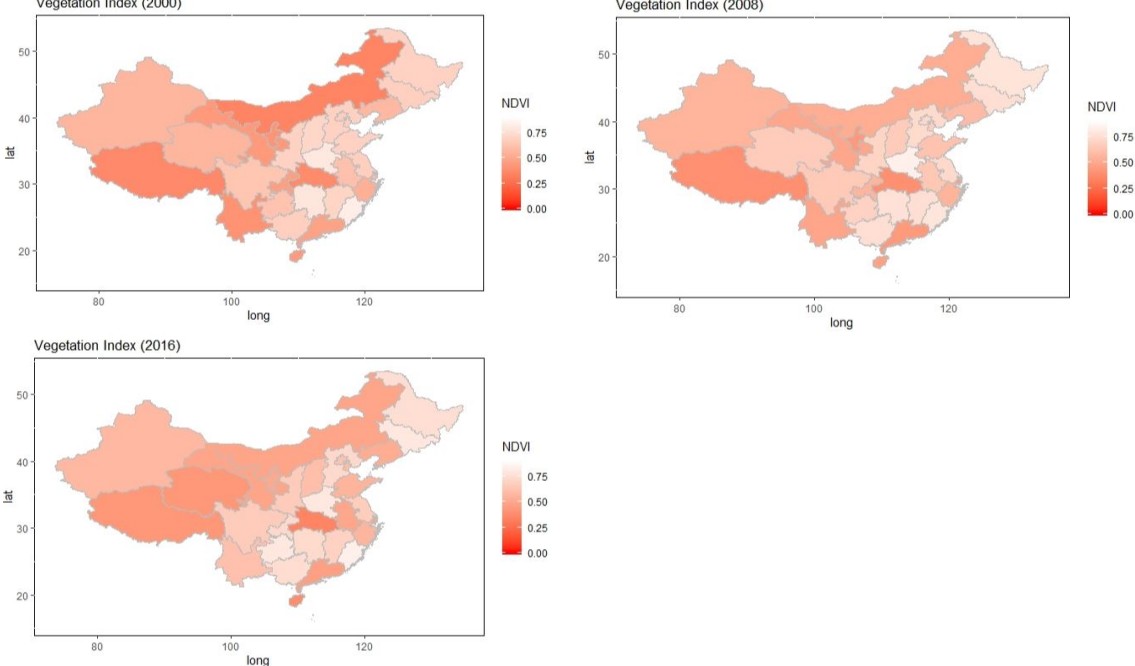

**Figure 7.** Normalized Difference Vegetation Index (NDVI) in different years in the Chinese territory (data in the Hong Kong, Macao, and Taiwan area are not indicated due to data unavailability).

**Table 2.** NDVI of all provincial capitals and its variation from 2000 to 2016.

| Provinces | Capital Cities | Location (Lat., Long.) (N, E) | NDVI '00 (-) | NDVI '08 (-) | NDVI '16 (-) | ΔNDVI ('08-'00) | ΔNDVI ('16-'08) | ΔNDVI ('16-'00) |
|---|---|---|---|---|---|---|---|---|
| Anhui | Hefei | 31.5°, 117.2° | 0.61 | 0.64 | 0.49 | 0.03 | −0.15 | −0.12 |
| Beijing | Beijing | 39.5°, 116.2° | 0.67 | 0.72 | 0.62 | 0.05 | −0.1 | −0.05 |
| Chongqing | Chongqing | 29.4°, 106.3° | 0.46 | 0.56 | 0.72 | 0.1 | 0.16 | 0.26 |
| Fujian | Fuzhou | 26.1°, 119.2° | 0.82 | 0.78 | 0.83 | −0.04 | 0.05 | 0.01 |
| Gansu | Lanzhou | 36.0°, 103.5° | 0.43 | 0.49 | 0.48 | 0.06 | −0.01 | 0.05 |
| Guangdong | Guangzhou | 23.1°, 113.1° | 0.47 | 0.43 | 0.46 | −0.04 | 0.03 | −0.01 |
| Guangxi | Nanning | 22.5°, 108.2° | 0.68 | 0.75 | 0.74 | 0.07 | −0.01 | 0.06 |
| Guizhou | Guiyang | 26.4°, 106.4° | 0.62 | 0.69 | 0.79 | 0.07 | 0.1 | 0.17 |
| Hainan | Haikou | 20.0°, 110.2° | 0.44 | 0.47 | 0.38 | 0.03 | −0.09 | −0.06 |
| Hebei | Shijiazhuang | 38.0°, 114.3° | 0.69 | 0.73 | 0.72 | 0.04 | −0.01 | 0.03 |
| Heilongjiang | Harbin | 45.4°, 126.4° | 0.69 | 0.77 | 0.75 | 0.08 | −0.02 | 0.06 |
| Henan | Zhengzhou | 34.5°, 113.4° | 0.79 | 0.83 | 0.79 | 0.04 | −0.04 | 0.00 |
| Hubei | Wuhan | 30.4°, 114.2° | 0.37 | 0.39 | 0.34 | 0.02 | −0.05 | −0.03 |
| Hunan | Changsha | 28.1°, 112.6° | 0.78 | 0.76 | 0.73 | −0.02 | −0.03 | −0.05 |
| Jiangsu | Nanjing | 32.0°, 118.5° | 0.68 | 0.72 | 0.67 | 0.04 | −0.05 | −0.01 |
| Jiangxi | Nanchang | 28.4°, 115.6° | 0.7 | 0.75 | 0.69 | 0.05 | −0.06 | −0.01 |
| Jilin | Changchun | 43.5°, 125.2° | 0.68 | 0.75 | 0.79 | 0.07 | 0.04 | 0.11 |
| Liaoning | Shenyang | 41.5°, 123.3° | 0.57 | 0.58 | 0.52 | 0.01 | −0.06 | −0.05 |
| Inner Mongolia | Hohhot | 40.5°, 111.4° | 0.34 | 0.52 | 0.48 | 0.18 | −0.04 | 0.14 |
| Ningxia | Yinchuan | 38.3°, 106.2° | 0.41 | 0.43 | 0.5 | 0.02 | 0.07 | 0.09 |
| Paracel Islands | Paracel Islands | 16.4°, 112.0° | 0.66 | 0.84 | 0.71 | 0.18 | −0.13 | 0.05 |
| Qinghai | Xining | 36.4°, 101.5° | 0.56 | 0.66 | 0.42 | 0.1 | −0.24 | −0.14 |
| Shaanxi | Taiyuan | 37.5°, 112.3° | 0.69 | 0.7 | 0.66 | 0.01 | −0.04 | −0.03 |
| Shandong | Jinan | 36.4°, 117.0° | 0.69 | 0.62 | 0.55 | −0.07 | −0.07 | −0.14 |
| Shanghai | Shanghai | 31.1°, 121.3° | 0.58 | 0.46 | 0.48 | −0.12 | 0.02 | −0.1 |
| Shanxi | Xian | 34.2°, 108.6° | 0.71 | 0.66 | 0.6 | −0.05 | −0.06 | −0.11 |
| Sichuan | Chengdu | 30.4°, 104.0° | 0.63 | 0.65 | 0.66 | 0.02 | 0.01 | 0.03 |
| Tianjin | Tianjin | 39.0°, 117.1° | 0.68 | 0.67 | 0.59 | −0.01 | −0.08 | −0.09 |
| Xinjiang | Urumqi | 43.5°, 87.4° | 0.56 | 0.55 | 0.57 | −0.01 | 0.02 | 0.01 |
| Tibet | Lhasa | 29.4°, 91.1° | 0.36 | 0.39 | 0.42 | 0.03 | 0.03 | 0.06 |
| Yunnan | Kunming | 25.0°, 102.4° | 0.4 | 0.48 | 0.61 | 0.08 | 0.13 | 0.21 |
| Zhejiang | Hangzhou | 30.2°, 120.1° | 0.53 | 0.55 | 0.56 | 0.02 | 0.01 | 0.03 |
| | Mean | | 0.592 | 0.625 | 0.604 | 0.033 | −0.021 | 0.012 |

(*NDVI* = Normalized Difference Vegetation Index; Δ*NDVI* = NDVI variation).

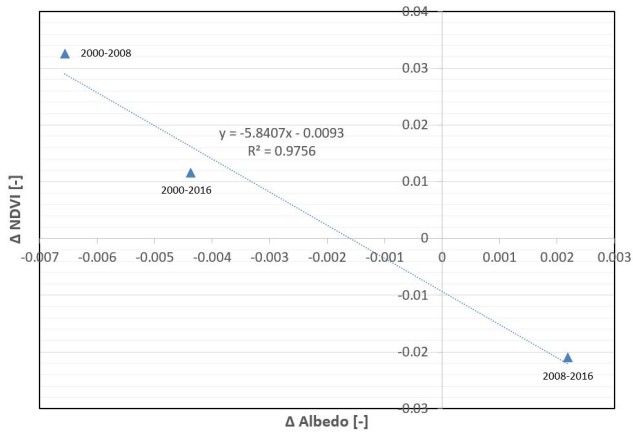

**Figure 8.** Correlation between temporal changes in NDVI and surface albedo.

## 4. Discussion

During the period from 2000 to 2008, the development of urbanization was rapidly carried out in Beijing and its surrounding regions (North China) [39], and light-colored building materials with relatively higher reflectivity were mainly used for the urban construction. Thus, it was considered that the surface albedo in North China, such as Beijing, increased most compared to the other regions of China. However, during this period, large-scale afforestation activities were also carried out in the Northeast and Northwest China, such as Jilin, Liaoning, and Inner Mongolia. As the forests had a low albedo, the surface albedo in these regions decreased the most, compared to the other regions of China.

Similar to the urbanization of Beijing and the surrounding regions from 2000 to 2008, the country began to focus on the urbanization in the Northeast and Northwest China, during the period of 2008 to 2016. Compared to the forest greening, the urbanization was more rapid in these regions. Thus, it was considered that the surface albedo increased most in the Northeast and Northwest China.

However, the surface albedo still showed a downward trend in the Northeast and in parts of the Northwest and Southwest China, from 2000 to 2016, even though forest greening and urbanization took place together. The reason was considered to be that the greening environment was more important than the urbanization from the overall planning view of recent years. The surface albedo in Guangdong, Fujian, and Shanghai and their surrounding regions showed relatively slight change, because the urbanization began in the nineteenth Century and the urbanization slowed down in the twentieth century for these regions.

A study by Tang et al. [22] reported that when the urbanization increased from 15% to 48.4%, it would lead to a decrease in urban albedo of approximately 0.05. However, it did not indicate what kinds of materials were used for urban construction. It was considered that the surface reflectivity of the building envelope material would strongly affect the urban albedo [40]. In addition, the study by Du et al. [41] indicated that urbanization has negative effects on NDVI, for most cities of China, over the last four decades. However, the effects of urbanization on NDVI showed obvious spatial differences. It was shown that urbanization generally resulted in the decrease of vegetation coverage in built-up areas for most of China's central and eastern metropolises, while in Western China, which is usually restricted by natural conditions, urbanization usually improved vegetation coverage. Therefore, it was considered that understanding and deeply learning the impacts of urbanization on variations in surface albedo and NDVI has become an important topic at present.

The results were only obtained by averaging the database of representative capital cities and their surrounding regions of provinces, not all cities of provinces, thus it might also cause great deviation in calculation in this study.

## 5. Conclusions and Future Work

In this study, based on the database of the MODIS instrument aboard NASA's Terra satellite, we investigated and analyzed the spatial and temporal land surface albedo variations for the entire Chinese territory, from 2000 to 2016. The surface albedo in the entire Chinese territory was classified using five different classification methods. Furthermore, the correlation between surface albedo and NDVI was also analyzed.

Northeast China (Jilin and Liaoning) showed the largest surface albedo decrease, about 0.18 (2008–2000) and about 0.09 (2016–2000), due to the green forest policy. North China (Beijing and Hebei) showed the largest surface albedo increase of about 0.07 (2008–2000 and 2016–2000), as the urbanization was more rapid. The surface albedo in Guangdong, Fujian, and Shanghai and their surrounding regions showed a relatively slight change, because the urbanization began in the nineteenth century and the urbanization slowed down in the twentieth century, for these regions. Furthermore, the analysis of the change in NDVI and surface albedo showed that the change trend of surface albedo was opposite to that of NDVI.

The land surface albedo was regarded to be an important indicator, which was related to the formulation of building energy conservation indictors that changed with urban climate change. In order to draw a map with higher resolution and find out the specific reasons for the change of surface albedo and NDVI, future research should be focused on the resolution of the sensor and analyzing the relationship between various factors, such as changes in ground cover and progress of urbanization, etc.

In addition, urban forms were regarded as an important factor that is strongly related to the UHI effect and land surface albedo. Previous research proved that urban forms affect urban microclimate [42], and changes in the urban forms result in modified building energy consumption [43]. Previous studies have proposed that by examining the relationship between urban forms and climate, one could employ the results of urban climatology into urban design guidelines [44,45]. Therefore, the research of urban forms should be systematically studied for exploring the relationship with changes in land surface albedo in the future.

**Funding:** This research received no external funding.

**Conflicts of Interest:** The author declare no conflict of interest.

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
