# Peer review of "Investigation of Spatial and Temporal Changes in the Land Surface Albedo for the Entire Chinese Territory"

_geosciences, doi:10.3390/geosciences10090362_

Round 1
Reviewer 1 Report
Thank you for your interesting research. I agree that albedo is very important factor for urban climate. However, I cannot understand what you want to highlight. The method you used cannot make a high resolution albedo map which can be used for identifying the relationship between urbanization and albedo.
Please see the attached file.

Author Response
Thank you for your interesting research. I agree that albedo is very important factor for urban climate. However, I cannot understand what you want to highlight. The method you used cannot make a high resolution albedo map which can be used for identifying the relationship between urbanization and albedo.
Response: Thank you for your comments. As you know, high-resolution satellite equipment similar to Google earth software can’t be applied to the map of China’s territory, thus the change in any land surface can’t be observed and mapped by high-resolution. Also, the map resolution is based on the accuracy of database, the MODIS resolution is 3600 x 1800 with 0.1 degrees in this study. If the accuracy of the database is improved, the accuracy of the analyzed mapping can be also improved. This study introduced using simple technology of drawing contour maps to investigate and analyze the change in surface albedo of China for better understanding the spatial and temporal variation of surface albedo in the whole China territory, based on the measurement database from the MODIS instrument aboard NASA’s Terra satellite. A higher resolution albedo map will be improved in the next work.
In addition, in the urban climate aspect, I want to know the meaning of increased (or decreased) urban albedo. Is it good or bad for heat mitigation?
Response: Thank you for your comment. Albedo is the ratio of reflected light to total incident sunlight for a given area of the land surface. It is a fundamental property controlling the energy flux at the surface of the Earth, which means it is an important parameter for climate and weather models. It can provide information on biophysical characteristics of the land surface including the structure of vegetation canopies, soil moisture, and urbanization. In the field of urban heat island mitigation strategies, it is considered that the greater the urban albedo, the better the urban heat islands mitigation.
Line 43: However, quantification of this effect on the global scale is difficult.
- I cannot agree with this sentence. Remote sensing technique already allows us to quantify the changes of albedo.
Response: Thank you for your comment. I strongly agree with what you said. I’m sorry that this sentence made you misunderstand the true meaning of this sentence. What I want to express is that apart from remote sensing technology, there are few simpler ways to popularize applications. This sentence has been revised accordingly.
- Aren’t there previous studies which quantify the relation between human activity and albedo?
Response: Thank you for your comment. As you said, there are some research in this field, however in terms of methods, there are remoting sensing technique and surface energy balance mechanism analysis technology. Also, these research has been cited in section “1.3 Urban albedo…”, such as reference Hu et al [19] and Hou et al [20], and so on.
Line 49-53: please put references for each sentence.
Response: Thank you for your advice. Added the reference accordingly.
Line 54: urban vegetation could reduce heat risk by generating evapotranspiration latent heat. What is the relationship between pollutant and UHI health impact? I hope you can exactly point out the reason why vegetation can reduce urban heat with various references.
Response: Thank you for your advice. Revised this sentence and added new reference.
Line 56, 60: the methods of referring are different. Please check the reference system.
Response: Thank you for your advice. Checked the reference again.
Line 61-67: I cannot understand the direction of these references. Some said the difference depending on height and some said the difference depending on areas / seasonal variation… If you want to highlight the ‘worldwide studying’ (line 56), you can just compare the urban heat island intensity of various worldwide cities and China.
Response: Thank you for your valuable comments. This section illustrates that the research on the urban heat island phenomenon is carried out all over the world. They are all comments on tropical phenomenon. In fact, these comments were added at the suggestion of other reviewers. I hope you can accept this explanation.
Line 69-70, 78: please put references.
Response: Thank you for your advice. Added the reference accordingly.
Line 75: The average albedo of the Earth at the top of the atmosphere, is 0.30-0.35 because of cloud cover
- You are talking about land surface albedo. and you do not need to say this atmosphere albedo.
Response: Thank you for your valuable comment. This sentence wants to show that the satellite technical observations of albedo are different in observation height, and the observed result is often affected by atmosphere. The similar sentence has been added to better understand.
Line 82-: The results showed there has been rapid urban expansion over the last decade, with the urban land area increasing at about 3.3 % annually from 2001 to 2009, resulting in lower albedo due to complex building configurations of urban centers and higher albedo on flat surfaces of suburban areas and cropland.
- Why urban albedo is lower than others? Because road pavement has dark color?
Response: Thank you for your valuable comment. Yes, as you said, with the development of the urbanization during 2001 to 2009, the road (asphalt) with low reflectivity were built and replaced the natural ground coverage with little higher reflectivity, so the urban albedo is becoming lower.
Line 89: Many studies indicated that urban temperatures and photochemical reaction 90 rates decreased with the increase in urban albedo
- Here, is albedo increasing for urbanization? It is so confused.
Response: Thank you for your valuable comment. I’m sorry I made a mistake because of negligence. The corrected sentence is “….. increased with the increase in urban albedo”. I have corrected it accordingly.
Line 91-93: I think this reference is not suitable.
Response: Thank you for your valuable comment. Removed unsuitable reference accordingly.
Line 68: There is no introduction about satellite technology. What is the research gap in the satellite technology?
Response: Thank you for your valuable comment. The content described in section 1.3 is directly related to satellite technology, however the term “satellite technology” didn’t appear. In order to better reflect, the word “satellite technology” has been added in this section.
Line 97: timing change-> temporal change
Response: Thank you for your valuable comment. Revised “timing change” to “temporal change” accordingly.
Line 101-103: why do we need the simple technology of drawing contour map? Maybe we can find higher resolution (raster file, not a contour) of global scale albedo. Wasn’t there any literature which quantified the spatial and temporal variation of China territory albedo? you should write this issue in the introduction.
Response: Thank you for your valuable comment. This research is to avoid the complexity of using other satellite technology analysis methods, and proposes different method of using the database to analyze itself. This may be more useful when high-precision analysis software can’t be used without research funding. Also, the reference [23] is a research that the spatial and temporal variation of albedo in Beijing region have been quantified using Landsat images of satellite technology, but it does not involve the whole China territory. The similar sentence has been added in the section 1.4.
Fig. 1: Please make a new map.
Response: Thank you for your valuable comment. Changed the map of China and revised the sentence accordingly.
Line 120: Why did you select only August?
Response: Thank you for your valuable comment. Because the ice or snow of Earth’s surface in winter would strongly affects the surface albedo, and to avoid these effect of ice and snow, the summer month (August) when there is no snow or ice in the whole China area was selected in this study. The similar explanation has been stated in the section 2.2.
Did you use MODIS product or raw MODIS image and calculate albedo?
Response: Thank you for your valuable comment. The MODIS database of csv file type with an interval of 0.1 degrees for the China territory were used for analyzing the surface albedo in this study.
2.3. Method of drawing contour maps?
- What is the strongpoint of contour maps compared to raster map?
Response: Thank you for your valuable comment. Compared to raster map that reflects the distribution of albedo, the contour map can also reflect the correlation and overall trends of albedo. Moreover, want to be different from the raster map often used for analysis, the contour map is used in this study.
- Why does the contour map suit for your research aim?
Response: Thank you for your valuable comment. Because the aim of this study is to investigate the spatial and temporal changes in the land surface albedo for the whole China territory, I think the contour map is more suitable for this study, compared to raster map.
Line 141: Do we need to compare the five classification methods? And why do you select these five methods?
Response: Thank you for your valuable comment. Classification (Figs.3-5) is the process of learning a model that can classify the surface albedo of all area with similarities between objects, which is groups according to those characteristics in common and which differentiate them from other groups of objects. And these groups are known as “clusters”. There are main five different classification methods often used, thus we applied five different classification methods to classify the surface albedo with different years in this study. In addition, the appropriate number of zones has been decided by different classification methods, thus there are 5 zones and 6 zones in this study. Classification just represents the areas with similar characteristics. In this study, I think it is not necessary to compare the five classification methods.
Methods: please write the method of comparison between albedo and NDVI. (not in a result part).
Response: Thank you for your valuable comment. Revised and moved the NDVI part to new section 2.5 accordingly.
Fig. 8. I think Fig 8 should be written in the result part. And why don’t you use the province data or raster value (not a representative data?) for calculating the correlation between surface albedo and NDVI?
Response: Thank you for your valuable comment. Moved the Fig.8 to the result section “3.5 Change in NDVI”. I’m sorry for my lack explanation, this study used the average surface albedo and NDVI of the provincial capital city and its surrounding areas for analysis, not a representative data.
I am not sure that the word ‘greening’ is really representing ‘NDVI’. I prefer to use ‘NDVI’ instead of ‘greening’.
Response: Thank you for your valuable comment. Corrected “greening” to “NDVI” accordingly.
There is no interpretation of relationship between albedo and NDVI. What is the meaning of the negative relationship?
Response: Thank you for your valuable comment. The negative value of Δalbedo and ΔNDVI represents a decrease in albedo and NDVI during the years. Thus, Fig.8 indicated the change trend of surface albedo is opposite to that of NDVI.
Line 272: I cannot understand the meaning.
Response: Thank you for your valuable comment. I’m sorry that this sentence confuses you, so I removed it and revised this sentence for better understand.

Reviewer 2 Report
The manuscript deals with a topic of important and certain interest. A simple and original method is proposed for the classification of the superficial albedo for the Chinese territory.
The organization is smooth and extremely legible even for a non-specialist public. The introduction is complete and the bibliographical references very precise.
The methodology is described very clearly and convincingly.
The description of the results is complete as regards the proposed text but consideration will follow.
The discussion develops coherently with the data collected, here too a consideration will follow.
Considerations:
1. Although the results obtained are clear, the reader is required to have some effort in positioning them correctly in time and space. Perhaps a graph that clusters the trends of values ​​over time for the different territories could be of valid help in fully understanding the results.
2. the discussion flows coherently up to page 11, lines 275-285, after which some pre-conclusions are introduced that do not bind directly to the data obtained. The two paragraphs in question are absolutely correct in content, but perhaps a subtitle would be needed to make the reader understand correctly that these considerations are of a more general nature. Or include this part of the text as a preamble to the conclusions.
The manuscript is therefore scientifically valid and well suited to the purposes of the Journal. I, therefore, suggest the major modifications option to let the authors evaluate the expressed considerations and include them in the review.
Author Response
The manuscript deals with a topic of important and certain interest. A simple and original method is proposed for the classification of the superficial albedo for the Chinese territory.
The organization is smooth and extremely legible even for a non-specialist public. The introduction is complete and the bibliographical references very precise.
The methodology is described very clearly and convincingly.
The description of the results is complete as regards the proposed text but consideration will follow.
The discussion develops coherently with the data collected, here too a consideration will follow.
Response: Thank you for your positive comments.
Considerations:
- Although the results obtained are clear, the reader is required to have some effort in positioning them correctly in time and space. Perhaps a graph that clusters the trends of values ​​over time for the different territories could be of valid help in fully understanding the results.
Response: Thank you for your valuable comment. I strongly agree with what you said. It is important to allow readers to better understand the spatial and temporal changes in the land surface albedo and NDVI. However, there are too many graphs in this study, so tables are used to reflect the spatial and temporal changes in the land surface albedo and NDVI.
- the discussion flows coherently up to page 11, lines 275-285, after which some pre-conclusions are introduced that do not bind directly to the data obtained. The two paragraphs in question are absolutely correct in content, but perhaps a subtitle would be needed to make the reader understand correctly that these considerations are of a more general nature. Or include this part of the text as a preamble to the conclusions.
Response: Thank you for your valuable comment. As you said, the two paragraphs seem to be very independent and have nothing to do with the discussion in the paper. Therefore, these two paragraphs have been revised and moved to the last section “Conclusions and future work” for better understand.
The manuscript is therefore scientifically valid and well suited to the purposes of the Journal. I, therefore, suggest the major modifications option to let the authors evaluate the expressed considerations and include them in the review.
Response: Thank you for your positive comment. I have revised this paper in accordance with reviewers’ valuable comments, and hope to meet your requirements. Thank you very much.

Round 2
Reviewer 1 Report
Thank you for your consideration of my comments. I can understand the reason why you select the method and input data for making albedo map for China.
There are two major comments.
1. You explained the reason you used the contour map. It was for comparing spatial and temporal variance. Here, please explain more about it in the introduction and method regarding some previous literature. I think you need refer other studies for claiming your opinion.
2. I suggest you that explain the meaning of Fig. 8. you wrote that 'The result indicated that the change trend of surface albedo is opposite to that of
NDVI' in the results. I agree that this sentence is enough for the result section. But my intend of 'explanation' is that writing the internal meaning in the discussion section. Please discuss about the relation between albedo and NDVI regarding urbanization, land cover change phenomenon.
Author Response
Thank you for your consideration of my comments. I can understand the reason why you select the method and input data for making albedo map for China.
Response: Thank you for your understanding and positive comment.
There are two major comments.
- You explained the reason you used the contour map. It was for comparing spatial and temporal variance. Here, please explain more about it in the introduction and method regarding some previous literature. I think you need refer other studies for claiming your opinion.
Response: Thank you for your valuable advice. Have added new references for better claiming the opinion of using contour maps in the introduction section accordingly.
- I suggest you that explain the meaning of Fig. 8. you wrote that 'The result indicated that the change trend of surface albedo is opposite to that of
NDVI' in the results. I agree that this sentence is enough for the result section. But my intend of 'explanation' is that writing the internal meaning in the discussion section. Please discuss about the relation between albedo and NDVI regarding urbanization, land cover change phenomenon.
Response: Thank you for your valuable advice. New reference on relation between albedo and NDVI have been added in discussion section, and corresponding discussions have also elaborated accordingly.
Reviewer 2 Report
The improvements performed by the authors are perfectly in line with the comments. I understand the problem in adding a new figure.
I suggest now the publication of the manuscript in the present new version
Author Response
The improvements performed by the authors are perfectly in line with the comments. I understand the problem in adding a new figure.
I suggest now the publication of the manuscript in the present new version
Response: Thank you for your positive comment of this paper.